# Effects of Red Ginseng Byproducts on Rumen Fermentation, Growth Performance, Blood Metabolites, and mRNA Expression of Heat Shock Proteins in Heat-Stressed Fattening Hanwoo Steers

**DOI:** 10.3390/vetsci9050220

**Published:** 2022-04-30

**Authors:** Daekyum Yoo, Hanbeen Kim, Joonbeom Moon, Jongnam Kim, Hyeran Kim, Jakyeom Seo

**Affiliations:** 1Department of Animal Science, Life and Industry Convergence Research Institute, Pusan National University, Miryang 50463, Korea; fbrjsgud@pusan.ac.kr (D.Y.); khb3850@pusan.ac.kr (H.K.); mantis0044@pusan.ac.kr (J.M.); 2Department of Food and Nutrition, Dongseo University, Busan 47011, Korea; yorker20@gdsu.dongseo.ac.kr; 3Animal Nutrition and Physiology Team, National Institute of Animal Science, RDA, Jeonju-si 55365, Korea; ococ1004@korea.kr

**Keywords:** red ginseng byproduct, heat stress, hanwoo steers, heat shock proteins

## Abstract

The objective of this study was to evaluate the effects of dietary supplementation with red ginseng byproduct (RGB) on rumen fermentation, growth performance, blood metabolites, and mRNA expression of heat shock proteins (HSP) in fattening Hanwoo steers under heat stress. Two experimental total mixed rations (TMR) were prepared: (1) a TMR meeting the requirement of fattening beef having an average daily gain (ADG) 0.8 kg/day (CON) and (2) a TMR that included 2% RGB on a dry matter (DM) basis (GINSENG). *In vitro* rumen fermentation and *in vivo* growth experiments were conducted using two experimental diets. A total of 22 Hanwoo steers were distributed to two treatments (CON vs. GINSENG) in a completely randomized block design according to body weight (BW). The experiment was conducted during the summer season for five weeks. The final BW, ADG, DM intake, and feed conversion ratio did not differ between treatments in the growth trial. In the mRNA expression results, only HSP 90 showed an increasing tendency in the GINSENG group. The use of 2%DM RGB did not improve the growth performance or alleviate heat stress in fattening Hanwoo steers during the summer season.

## 1. Introduction

The occurrence of heat stress (HS) in domestic animals due to climate change has become a critical issue in the livestock industry. The South Korea Meteorological Agency reported that surface air temperatures (AT) will likely increase by 6 °C across the Korean Peninsula by the end of the twenty-first century [1]. In Korea, the summer period is characterized by high temperature and high humidity conditions that may induce HS in animals. Oxidative stress induced by HS negatively affects cell and tissue damage, nucleic acid destruction, protein denaturation, and antioxidant enzyme activity, leading to health problems in animals [2,3]. Under HS conditions, ruminants exhibit decreased feed intake, decreased rumination, and impaired nutrient absorption; therefore, the nutrient requirements for maintaining their body weight (BW) increase [4]. Several studies have suggested management strategies to alleviate HS, such as shade utilization, cooling systems and selection of heat-tolerant cattle [5].

The biological mechanisms that mediate physiological responses under HS are not well understood; however, the changes in metabolic markers regarding immune response [6] or oxidative stress [7] under HS conditions indicate that heat-stressed animals are highly sensitive to inflammatory responses. Therefore, the use of bioactive compounds to mitigate excessive immune responses and oxidative stress may improve animal production under HS conditions. It has been reported that ginsenoside Rh1 in ginseng inhibits the expression of oxidative stress and pro-inflammatory cytokines, whereas it increases the expression of anti-inflammatory interleukin 10 (IL-10) [8]. Red ginseng has also been reported to reduce HS in mice [9]. However, to our knowledge, little research has been conducted on the use of ginseng or ginseng byproducts as feed ingredients to mitigate HS in beef steers.

Red ginseng byproduct (RGB) was the residue after a boil-steaming process to extract water soluble and biologically active compounds from red ginseng [10]. Hong et al. [11] reported that the RGB still has a large amount of active ingredients (polysaccharide, ginsenoside, etc.) which were not insoluble in water. Kim et al. [12,13] observed that RGB contained ginsenoside Rb1 (8.97 μg/mL), ginsenoside Rg3 (15.65 μg/mL), and acidic polysaccharides (12.10 mg/mL). Therefore, this study investigated whether the use of RGB in conventional diets could improve the growth performance in heat-stressed fattening Hanwoo steers.

## 2. Materials and Methods

The protocols for animal use in this study were reviewed and approved by the Animal Research Ethics Committee of Pusan University (PNU-2020-2827).

### 2.1. Preparation of Experimental Diets and Chemical Analysis

The main ingredients for two experimental total mixed rations (TMR) were a commercial concentrate mix (Nonghyup Feed, Co., Ltd., Miryang, Korea), timothy hay, corn flake, molasses and RGB. The TMR without RGB was used as the control diet (CON), whereas the treatment TMR (GINSENG) contained 2%DM RGB, replacing 2%DM timothy hay as the chemical composition of RGB that is similar to that of timothy hay (Table 1). The feed ingredients and chemical composition are listed in Table 2. Both diets were formulated to meet or exceed the average daily gain (ADG) of 0.8 kg/d following nutrient requirements for beef cattle [14]. All feed ingredients and trial diets were dried at 65 °C for 72 h before being ground in a cyclone mill (Foss Tecator Cyclotec 1093, Foss, Hillerød, Denmark) with a 1 mm screen. Dry matter (DM) was assessed using the National Forage Testing Association method 2.2.2.5 [15] and ash (#942.05), acid detergent fiber (ADF, #973.18), and crude protein (CP, #990.03) were analyzed using the methods of the Association of Official Analytical Chemists International [16]. The CP content was calculated by multiplying the nitrogen content by 6.25. Total nitrogen was determined according to the Kjeldahl method using a nitrogen combustion analyzer (Leco FP-528, Leco, MI, USA). The fiber content of neutral detergent fiber (aNDF) and acid detergent lignin was determined using the procedures published by Van Soest et al. [17]. The aNDF was estimated using heat-stable amylase (α-amylase) and expressed with residual ash included. The total digestible nutrients and net energy required to maintain the experimental diets were estimated using the formulae [14,18].

### 2.2. In Vitro Fermentation

*In vitro* fermentation was undertaken using rumen fluid collected from two cannulated Holstein steers (BW: 680 ± 30 kg) before morning feeding at the Center for Agriculture Research, Pusan National University, Korea. In total, *in vitro* fermentation was performed two times. Animals were fed fattening phase TMR (Nonghyup Feed Co., Ltd. Miryang, Korea) twice a day. Rumen fluid was collected 1 h before the morning feeding time, mixed, transferred into a thermos bottle and immediately transported to the laboratory. The rumen contents were filtered through eight layers of cheese cloth and mixed with 4× volume of *in vitro* rumen buffer solution [19], which was modified under strictly anaerobic conditions. In pre-weighed nylon bags (R55, ANKOM Technology, Macedon, NY, USA), around 0.5 g of the pulverized experimental substrates were inserted. All bags were heat-sealed and transferred into empty 125 mL serum bottles. Four bottles were used for each dietary treatment and each bottle contained two bags. Following that, 70 mL of rumen fluid and buffer mixture was transferred (pH: 6.70, Temperature: 39 °C), along with continual flushing withO_2_-free CO_2_ gas. The bottles were sealed with butyl rubber stoppers and aluminum caps and incubated on a rotary shaker (JSSI-300T, JS Research Inc., Gongju, Korea) at 20 rpm for 48 h at 39 °C. DM digestibility (IVDMD), CP digestibility (IVCPD) and aNDF digestibility (IVNDFD), pH, ammonia nitrogen (NH_3_-N), volatile fatty acids (VFA) concentrations and gas production were determined after 48 h of incubation. Gas production was determined at 3, 6, 12, 24, 36, and 48 h using a pressure transducer (Sun Bee Instrument Inc., Seoul, Korea), as described previously [20]. Gas production profiles obtained during incubation were fitted to a simple exponential model, the equation for which is as follows [21]:*V_T_* = 0   (0 ≤ *T* ≤ *L*)
*V_T_* = *V_max_* × {1 − *e*^[−*kg* × (*T* − *L*)]^}  (*T* ≥ *L*)
where *T* is the time (h), *L* is the lag time (h), *e* is the exponential function, *k_g_* is the fractional rate of gas production (h^−1^), *V_T_* is the gas produced at time *T* (mL), and *V_max_* is the theoretical maximum gas production (mL) after the asymptote is reached. The values of all metabolites and gas production in blank bottles were used to correct data for samples. The bottle caps were removed after incubation and the bottles were immediately placed on ice to stop the fermentation. The bottles’ nylon bags were removed and cleaned with water until the water ran clear. The washed bags were dried at 65 °C for 72 h and weighed to measure the IVDMD. The aNDF content of the weighed bags was assessed using a modified version of the micro-NDF method proposed by Pell and Schofield [22] to evaluate IVNDFD. Kjeldahl nitrogen analysis was used to calculate IVCPD. The remaining culture fluid (approximately 50 mL) was transferred to a centrifuge tube and centrifuged at 2,500× *g* for 20 min at 4 °C. The supernatant was collected to determine the pH, VFA concentration, and NH_3_-N concentration. The pH of the culture fluid was measured using a pH meter (FP20, Mettler Toledo, OH, USA). The supernatant (1 mL) for VFA analysis was acidified with 200 μL of 25% meta-phosphoric acid, whereas the supernatant (1 mL) for NH_3_-N analysis was acidified with 200 μL of 0.2 M sulfuric acid and both were stored at −80 °C until VFA and NH_3_-N analysis. For VFA analysis, 200 μL of the supernatant was diluted with 800 μL of anhydrous ethyl alcohol (4023-2304, Daejung Chemicals, Siheung, Korea) after 15 min of centrifugation at 20,000× *g*. A gas chromatograph (Agilent 7890A, Agilent Technology, Santa Clara, CA, USA) with a flame ionization detector and capillary column (NukolTM fused silica capillary column, 30 m × 250 μm × 0.25 μm, Supelco Inc., Bellefonte, PA, USA) was used to determine the VFA content. The temperatures of the oven, injector and detector were set at 90 °C, 90 to 200 °C (rate: 15 °C/min; hold time: 2 min), and 230 °C (rate: 20 °C/min; hold time: 8 min), respectively. At a flow rate of 30 mL/min, nitrogen was used as the carrier gas. The NH_3_-N concentration was analyzed with several modifications [23]. After samples were centrifuged for 15 min at 20,000× *g* at 4 °C, 2 μL of the supernatant was mixed with 100 μL of alkali hypochlorite and phenol color reagent. After that, the mixture was incubated for 15 min at 39 °C in a water bath. Using a microplate reader (iMark, Bio-Rad, Hercules, CA, USA), the optical density at 630 nm was measured to determine the NH_3_-N concentration.

### 2.3. In Vivo Experimental Design

The species (*Bos taurus Coreanae*) of cattle used in the *in vivo* trial was Korean beef cattle. A total of 22 Hanwoo steers (21 ± 0.5 months old; BW: 571 ± 39.5 kg) were distributed to two treatments (CON vs GINSENG) in a completely randomized block design according to BW. Each group had three replicate pens (5 m × 5 m) and four, four, and three steers of similar BW were placed together in each pen. Each pen was equipped with an automated water drinker. All diets were manufactured weekly and fed to the cattle twice per day (8:00 a.m. and 3:00 p.m.) ad libitum (targeting 10% refusal). The experiment was conducted during the summer season for five weeks, with the first week as the adaptation period. The daily DM intake (DMI) was determined by measuring the feed offered and refused in each pen daily. All steers were weighed on the first, 14th, and 28th days of the experiment to determine their feed conversion ratio (FCR) and ADG.

### 2.4. Temperature–Humidity Index

The AT (°C) and relative humidity (RH, %) were recorded three times per day using 2 temperature–humidity meters (AcuRite 01083M, AcuRite Co., Ltd., Seoul, Korea). The height of the temperature–humidity meter from the ground was 2.5 m. The temperature–humidity index (THI) was calculated using the following equation [24]: THI = 0.8 × AT + [RH × (AT − 14.4)] + 46.4.

### 2.5. Respiration Rate and Rectal Temperature

We measured respiration rate and rectal temperature twice weekly during the experimental period. The respiration rate was recorded manually from 3:00 p.m. to 5:00 p.m. (using a stopwatch and counting uninterrupted flank movement: breathing for 1 min). The rectal temperature was recorded manually between 3:00 p.m. and 5:00 p.m. using a digital MT 200 thermometer (Microlife Inc., Clearwater, FL, USA).

### 2.6. Rumen Fermentation Characteristics

On the 13th and 27th days of the experiment, rumen fluid from all steers was collected using an oral stomach tube immediately before afternoon feeding and immediately transported to the laboratory. The rumen fluid was centrifuged at 2500× *g* for 20 min at 4 °C. The pH, VFA content, and NH_3_-N concentration were analyzed in the same manner as described above.

### 2.7. Analysis of Blood Metabolites

Blood was collected every two weeks during the experiment after afternoon feeding. The blood sample was collected from each steer’s jugular vein and placed in a serum tube containing a clot activator (BD Vacutainer; BD and Co., Franklin Lakes, NJ, USA). Blood was centrifuged at 2,500× *g* for 15 min at 4 °C in serum tubes. Blood parameters in serum including glucose (Glu), inorganic phosphate (IP), total cholesterol (T-Chol), triglycerides (TG), blood urea nitrogen (BUN), creatinine (Crea), albumin (Alb), aspartate aminotransferase (AST), alanine aminotransferase (ALT), total protein (TP), Mg and Ca were all measured in serum using Wako Pure Chemical Industries, Ltd. kits (Osaka, Japan). An automated chemistry analyzer was used to examine all parameters (Toshiba Acute Biochemical Analyzer-TBA-40FR, Toshiba Medical Instruments, Tokyo, Japan).

### 2.8. Total RNA Isolation and Real-Time Quantitative Polymerase Chain Reaction

Blood samples were collected from the jugular vein of the cattle in 10 mL K_2_-EDTA vacutainer tubes (BD Vacutainer, Becton Dickinson Co., Franklin Lakes, NJ, USA). Blood samples (10 mL) from each animal were immediately placed on ice and transported to the laboratory for the isolation of peripheral blood mononuclear cells (PBMC) within 30 min of sampling. PBMC were isolated by density gradient centrifugation. Briefly, whole blood samples were diluted with phosphate-buffered saline (PBS) at a 1:1 ratio in 15 mL conical tubes. Subsequently, 4 mL of Lymphoprep (STEMCELL Technologies Inc., Vancouver, BC, Canada) were added to the new tubes and 8 mL of the diluted blood samples were overlaid on the Lymphoprep. After centrifugation for 20 min at 800× *g* and 22 °C, the layer of cells above the Lymphoprep was collected and washed twice with PBS to obtain purified PBMC that were then suspended in 1 mL of TRIzol reagent (Invitrogen, Carlsbad, CA, USA) and transferred to a 1.5 mL tube. The PBMC were immediately stored at −80 °C until RNA isolation. RNA was isolated from the PBMC samples in TRIzol and the concentration and purity of total RNA were measured using a NanoDrop spectrophotometer (ND-1000, Thermo Fisher, MA, USA). For further examination, RNA having an OD value of greater than 1.8 (260/280 nm) was chosen and stored at −80 °C until the experiment was carried out.

First-strand cDNA was synthesized using 1 μg of RNA and 1 μL each of oilgodT (Invitrogen, Waltham, MA, USA) and AccuPower RT PreMix (Bioneer, Daejeon, Korea), according to the manufacturer’s instructions. Real-time quantitative polymerase chain reaction (PCR) assays were performed using a CFX96 Touch System (Bio-Rad Laboratories Inc., Hercules, CA, USA). Information on the primer sequences of cytokines and HSP was collected from previous studies (Table 3). The primers were designed using the National Center for Biotechnology Information Primer-BLAST (Table 3). Reactions were performed in triplicate with reaction volumes of 20 μL, using optical reaction plates sealed with optical adhesive film. The interleukin-1β (IL-1B), interleukin 6 (IL-6), IL-10, heat shock proteins (HSP) 70, and HSP 90 reaction mixture contained 0.5 μL of 10 mM dNTP Mix (BioFACT, Daejeon, Korea), 2.0 μL of 10 × buffer (BioFACT, Daejeon, Korea), 1.0 μL of cDNA (50 ng), 1.0 μL of forward primer (10 μM), 1.0 μL of reverse primer (10 μM), 0.2 μL of Taq polymerase (BioFACT, Daejeon, Korea), 1.0 μL of EvaGreen (SolGent Co., Ltd., Daejeon, Korea), and 13.3 μL of PCR grade water. The tumor necrosis factor-α (TNF-a) reaction mixture contained 0.5 μL of 10 mM dNTP Mix (BioFACT, Daejeon, Korea), 2.0 μL of hot start 10 × buffer (SolGent Co., Ltd., Daejeon, Korea), 1.0 μL of cDNA (50 ng), 1.0 μL of forward primer (10 μM), 1.0 μL of reverse primer (10 μM), 0.2 μL of hot start Taq polymerase (SolGent Co., Ltd., Daejeon, Korea), 1.0 μL of EvaGreen (SolGent Co., Ltd., Daejeon, Korea) and 13.3 μL of PCR grade water. Real-time PCR was performed according to the manufacturer’s instructions (Table 3). Fluorescence was recorded at the end of each denaturation and extension step and the specificity of the amplicon was confirmed via the dissociation curve analysis of PCR end products by increasing the temperature from 65 to 95 °C at a rate of 0.5 °C every 5 s. The threshold cycles for each sample were normalized to a housekeeping gene (β-actin) and the 2^−ΔΔCt^ method was used to determine the relative gene expression.

### 2.9. Statistical Analysis

All data were checked for normal distribution using the Shapiro–Wilk test in the SAS 9.4 (SAS Institute Inc., Cary, NC, USA) package. When the data did not show normal distribution, the data were transformed (log or square root) to have a normal distribution. Then, all the *in vitro* data were analyzed using *t*-tests in SAS 9.4 (SAS Institute Inc., Cary, NC, USA).

The data of *in vivo* experiment was analyzed using the PROC GLIMMIX procedure of SAS 9.4 (SAS Institute Inc., Cary, NC, USA) according to the following model:*Y_ij_* = *µ* + *R_i_* + *T_j_* + *E_ij_*
in which *Y_ij_* is the response variable, *µ* is the overall mean, *R_i_* is the random effect of the pen within block (*i* = 1 to 3), *T_j_* is fixed effect of treatment (*j =* 1 to 2), and *E_ij_* is the residual error. Differences among treatments were also compared with the Tukey’s test when there was a significant overall treatment effect. Statistical significance was declared at *p* < 0.05, and a trend was speculated at 0.05 ≤ *p* < 0.15.

## 3. Results

### 3.1. In Vitro Experiment

The effects of GINSENG on gas production, gas parameters, and *in vitro* fermentation characteristics are presented in Table 4. There was no difference in gas production between the treatments until 36 h of incubation. The gas production at final incubation (48 h) tended to be higher in the GINSENG group than in the CON group (*p* = 0.057). No significant differences were detected in *V_max_* and *K_g_* between the treatments. GINSENG supplementation significantly increased the IVDMD, IVNDFD, IVCPD, and NH_3_-N (*p* = 0.006, *p* = 0.002, *p* < 0.001 and *p* = 0.043, respectively). There were no differences in the VFA concentration and individual VFA proportions between treatments. The pH was significantly lower in the GINSENG group than in the CON group (*p* = 0.001).

### 3.2. In Vivo Experiment

The data for growth performance, respiration rate and rectal temperature of Hanwoo steers fed each experimental diet are shown in Table 5. There were no significant differences in the final BW, ADG, and FCR between treatments. In addition, GINSENG did not exhibit the significant difference in respiration rate and rectal temperature compared to CON.

The rumen fermentation characteristics from the *in vivo* trial are shown in Table 6. There were no significant changes in total VFA production, each VFA proportion, and pH between treatments, but the NH_3_-N concentration was significantly higher in the GINSENG group compared to CON (*p* = 0.003).

The comparison of blood metabolites from steers fed different diets are shown in Table 7. Compared to the CON group, GINSENG significantly increased the BUN and AST levels in the blood (*p* = 0.013 and *p* < 0.001, respectively). The ALT tended to be higher in the GINSENG group than in the CON group (*p* = 0.088). Differences in Ca, IP, Mg, TG, Alb, T-Chol, Glu, and Crea were not observed.

The quantification of mRNA expression regarding inflammatory cytokines and HSP in PBMC are shown in Figure 1. Steers fed GINSENG showed no significant changes in pro-inflammatory cytokines (TNF-a, IL-1B and IL-6), anti-inflammatory cytokines (IL-10) and HSP 70 in PBMC. However, HSP 90 tended to increase in GINSENG compared to CON (*p* = 0.119; Figure 1).

## 4. Discussion

### 4.1. In Vitro Experiment

Before performing the *in vivo* experiment, an *in vitro* experiment was conducted to evaluate the effect of RGB supplementation on rumen fermentation. In this study, the final gas production in GINSENG tended to be higher than CON. In addition, the IVDMD and IVNDFD of the GINSENG group were significantly higher than those of the CON group. To prepare the treatment diet, 2%DM of timothy in CON was replaced with RGB because of their similar chemical composition. RGB is a residue after water soluble contents were extracted from red ginseng via the boiling process. Therefore, the nutrients, especially fiber carbohydrates in RGB, may be more favorable for degradation than those in timothy hay, even though their chemical compositions are similar. RGB also contains abundant bioactive compounds that can modulate rumen microbial fermentation [30], thereby higher nutrient degradation in GINSENG might be possible. Contrary to our results, Kim et al. [31] reported that the replacement of alfalfa hay (0% to 15%) with ginseng meal did not result in increased IVDMD. This might be due to differences in the processing methods when ginseng meal is used. Because the ginseng meal used by Kim et al. [31] was not treated with steam, the degradability of ginseng meal might differ to RGB used in this study. GINSENG also had significantly higher IVCPD and NH_3_-N concentrations than the CON. The NH_3_-N concentration was positively correlated with IVCPD because the amino acids in the feed protein were degraded to the carbon skeleton and NH_3_ by ruminal microbes.

Based on the pH, gas production, IVNDFD, and IVDMD results, it was predicted that the total VFA concentration would be higher in the GINSENG treatment. However, the total VFA concentration between treatments did not differ in CON. Hamid et al. [30] reported that the IVDMD and aNDF digestibility of corn gluten feed and RGB were similar, but the total VFA production was significantly lower in RGB. It has also been reported that the pH of RGB is significantly lower than that of corn gluten feed. In addition to VFA, lactate, formate, succinate, and ethanol are produced by rumen microorganisms as the end products and lactate affects the decrease in rumen pH [32]. Considering that GINSENG showed higher nutrient degradability than CON in this study, it was presumed that the lower pH in GINSENG might have been due to unmeasured organic acids, except in VFA.

### 4.2. In Vivo Experiment

Throughout the experimental period, the average daily maximum THI was 83.1 ± 4.96 and the average daily minimum THI was 66.4 ± 2.93 (Table 8). Throughout the trial, the mean daily THI was above 68, which indicates HS exposure in cattle [33]. In a previous study, respiration rate and rectal temperature were used as physical markers for heat-stressed animals [34]. Ruminant also exhibited increased respiratory counts as the first visible sign under HS conditions [35]. The higher rectal temperature value than the normal cows (38.3 to 38.7 °C) indicated HS and insufficient thermoregulation [36]. In this study, the respiration rate and rectal temperature did not differ between treatments. The rectal temperature in all steers in both diet groups was greater than 38.7 °C, indicating that they experienced HS during the experimental period regardless of the feed treatment. Sandner et al. [37] reported that a ginseng extract containing the major ginsenosides Re, Rg1, Rc, Rb2, and Rd might be a suitable feed additive to reduce the negative physiological effects under HS. RGB is a byproduct after the primary extraction of major bioactive substances from red ginseng; therefore, the effect of ginseng extract used in a previous study [37] might be higher than RGB to reduce negative effects by HS.

Based on the higher nutrient digestibility during *in vitro* tests in the GINSENG group, we speculated that steers fed GINSENG would exhibit improved growth performance. However, there were no significant differences in the final BW, DMI, ADG, and FCR. Colombo et al. [5] suggested that hyperthermia impacts growth performance by reducing feed intake and altering metabolic processes associated with feed efficiency. Considering the high values of HS indexes (THI and rectal temperature), steers in both groups were under HS conditions and the RGB supplementation did not alleviate the HS. Thus, no significant difference in growth performances between treatments might be explained partially.

The NH_3_-N is produced by the microbial fermentation of protein sources in feed. The NH_3_-N in the GINSENG (6.14 mg/dL) was higher than that in CON (3.56 mg/dL). Oh et al. [38] reported that the rumen NH_3_-N concentration of fattening Hanwoo steer was in the range of approximately 4–15 mg/dL immediately after feeding. Ruminal NH_3_-N in the current study was within the range (3.3 to 8.5 mg/dL) previously reported as optimum for rumen fermentation [39]. There was no significant difference in VFA production between dietary groups (CON: 65.7 mM; GINSENG: 66.1 mM). Kim et al. [40] observed that the VFA production in fattening Hanwoo generally ranged from 65–72 mM before feeding. Baek et al. [41] reported that the VFA production of Hanwoo steers was ranging from 68–82 mM immediately after feeding. Therefore, considering all of NH_3_-N, VFA production, each VFA proportions, and pH, it is presumed that the supply of RGB did not negatively affect the rumen fermentation.

It has been reported that there is a direct, positive relationship between the rumen NH_3_-N concentration and the BUN concentration because the rumen NH_3_-N is absorbed into the bloodstream [42]. According to the rumen fermentation characteristics, NH_3_-N was significantly higher in GINSENG and a similar result was observed *in vitro*. Therefore, the higher serum BUN in GINSENG might be linked with the ruminal NH_3_-N result. In a previous study, AST and ALT in serum were used as elements for the reliable hepatic diagnostics in animals [43]. The AST in the GINSENG was higher than that in CON (CON: 52.4 U/L; GINSENG: 70.0 U/L). ALT showed a higher tendency in the GINSENG than in the CON (CON: 17.3 U/L; GINSENG: 19.2 U/L). Previous studies had reported that the AST enzyme concentration range of 59.9–88.7 U/L and the ALT enzyme concentration range of 14.9–25.3 U/L in the fattening phase of Hanwoo steers [44,45,46]. In our study, both AST and ALT were within the range suggested by previous studies, so it is presumed that RGB did not have a negative effect on liver. Thus, considering blood metabolites, IVCPD, and NH_3_-N, it is assumed that the RGB diet increases proteolysis by rumen microorganisms and NH_3_-N increased by proteolysis seems to affect BUN concentration.

In this study, there was no significant change in pro-inflammatory cytokines (TNF-a, IL-1B and IL-6) between treatments. In addition, GINSENG did not affect the anti-inflammatory cytokine IL-10 compared with CON. Increased local and systemic reactive oxygen species and inflammatory responses induced by HS have been reported in livestock [47]. Inflammation is an immune response to harmful stimuli, such as pathogens, damaged cells, toxic compounds, or stress and acts by removing injurious stimuli and initiating the healing process [48]. Therefore, inflammation is a defense mechanism that is vital to health. Pro-inflammatory cytokines (IL-1B, IL-6, TNF-a, etc.) are involved in the induction of inflammation. Pro-inflammatory cytokines are mainly released by macrophages and are also produced in other tissues [49]. During inflammation, particularly in livestock, nutrients are spent in maintaining the immune response rather than in growing, ultimately decreasing the feed efficiency [47]. Kopalli et al. [50] noted that, when heat-stressed mice were fed 0.2% KGC04P (a component of red ginseng), the expression levels of TNF-a, IL-1B, and IL-6 decreased, thereby relieving HS. In this study, there was no inhibitory effect on the expression of pro-inflammatory cytokines (TNF-a, IL-1B, and IL-6) in the GINSENG group, thereby indicating that dietary RGB supplementation did not alleviate the inflammatory response in heat-stressed beef steers. In this study, the HSP 90 of GINSENG tended to increase compared with CON, although there was no difference in HSP 70. Yoon et al. [51] reported that, when Panax ginseng extract was supplied to heat-stressed rats, the mRNA expression of HSP 70 and HSP 90 in the liver decreased by 32% and 28%, respectively. Previous studies have reported that extracts containing ginsenosides reduce HS in rats and mice [9,52]. It is well known that the expression of HSP is involved in a cellular defense mechanism that enables cells to deal with stressful conditions, including HS, which induces oxidative stress and inflammation [53]. Deb et al. [54] reported that increased HSP 90 mRNA expression might be linked with body temperature maintenance and cell viability in beef cattle under HS condition. In this study, the increased tendency of HSP 90 mRNA expression in PBMC was observed in steers fed GINSENG, indicating that the energy to maintain a normal cellular defense mechanism might be more required than those fed CON; therefore, the RGB supplementation did not improve growth performance.

## 5. Conclusions

This study was conducted to investigate the efficacy of dietary RGB supplementation on rumen fermentation, growth performance, blood metabolites, and mRNA expression of inflammatory cytokines and HSP in heat-stressed fattening Hanwoo steers. In an *in vitro* experiment, higher nutrient degradability was observed in the GINSENG treatment, but, in the feeding trial, the supply of dietary RGB did not improve the growth performance. The expression of HSP 90 increased in steers fed GINSENG, whereas the expression of pro-inflammatory cytokines did not decrease. In conclusion, the use of 2% RGB did not improve the growth performance or alleviate HS in fattening Hanwoo steers during the summer season. Therefore, before applying the RGB as a feed ingredient to mitigate heat stress in beef cattle, the nutritive value of RGB (i.e., the amount of bio-active compounds, chemical composition, additional level and cost etc.) should be carefully considered.

## Figures and Tables

**Figure 1 vetsci-09-00220-f001:**
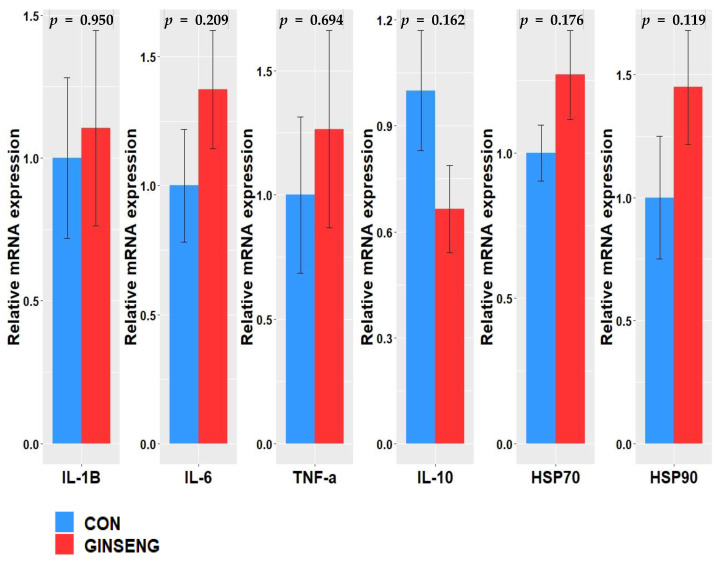
Differences in gene expression of inflammatory cytokine and heat shock protein in Hanwoo steers.

**Table 1 vetsci-09-00220-t001:** Chemical composition of red ginseng byproduct and timothy hay.

Items	RGB ^(1)^	Timothy Hay
Chemical composition		
DM (%as fed)	90.5	93.3
aNDF (%DM)	54.5	64.4
ADF (%DM)	47.4	42.6
Lignin (%DM)	14.7	6.60
Ash (%DM)	7.51	7.69
EE (%DM)	1.03	1.68
CP (%DM)	17.0	11.5
TDN (%DM)	44.3	54.8

DM, dry matter; aNDF, neutral detergent fiber analyzed with heat-stable α-amylase; ADF, acid detergent fiber; EE, ether extract; CP, crude protein; TDN, total digestible nutrients. ^(1)^ Red ginseng byproduct.

**Table 2 vetsci-09-00220-t002:** Experimental diet formulation and chemical composition.

Items	CON	GINSENG
Ingredients (% DM)		
Commercial concentrate mix	43.5	43.5
Corn flake	26.7	26.7
Timothy	26.2	24.2
Red ginseng byproduct	0	2.0
Molasses	3.1	3.1
Vitamin and mineral mix ^(1)^	0.5	0.5
Chemical composition		
DM (%as fed)	65.0	65.0
aNDF (%DM)	33.1	32.7
ADF (%DM)	20.3	20.3
Lignin (%DM)	3.20	3.40
Ash (%DM)	6.12	6.14
EE (%DM)	4.66	4.66
CP (%DM)	12.3	12.5
TDN (%DM)	70.7	71.0
NEm (Mcal/kg of DM)	1.67	1.66

DM, dry matter; aNDF, neutral detergent fiber analyzed with heat stable α-amylase; ADF, acid detergent fiber; EE, ether extract; CP, crude protein; TDN, total digestible nutrients; NEm, net energy for maintenance. ^(1)^ 33,330,000 IU/kg of vitamin A, 40,000,000 IU/kg of vitamin D, 20.86 IU/kg of vitamin E, 20.0 mg/kg of Cu, 90.0 mg/kg of Mn, 100.0 mg/kg of Zn, 250.0 mg/kg of Fe, 0.4 mg/kg of I and 0.4 mg/kg of Se.

**Table 3 vetsci-09-00220-t003:** Polymerase chain reaction primers used in this study.

Gene	Primer Sequences		Primer Condition [(initialization) → (Denaturation → Annealing → Elongation)]	Accession Number	Size (bp)	Reference
IL-1B	F	AGTGCCTACGCACATGTCTTC	one cycle (95 °C, 3 min) → 40 cycles (95 °C, 30 s → 60 °C, 30 s → 72 °C, 30 s)	NM_174093.1	114	[25]
	R	TGCGTCACACAGAAACTC GTC				
IL-6	F	CACCCCAGGCAGACTACTTC	one cycle (95 °C, 3 min) → 40 cycles (95 °C, 30 s → 64 °C, 30 s → 72 °C, 30 s)	NM_173923.2	215	
	R	AGCAAATCGCCTGATTGAAC				
IL-10	F	AAGGTGAAGAGAGTCTTCAGTGAGC	one cycle (95 °C, 3 min) → 40 cycles (95 °C, 30 s → 63 °C, 30 s → 72 °C, 30 s)	NM_174088	208	[26]
	R	TGCATCTTCGTTGTCATGTAGG				
TNF-a	F	GCTCCAGAAGTTGCTTGTGC	one cycle (95 °C, 10 min) → 40 cycles (95 °C, 30 s → 60 °C, 30 s → 72 °C, 30 s)	NM_173966.3	149	[27]
	R	AACCAGAGGGCTGTTGATGG				
HSP 70	F	TACGTGGCCTTCACCGATAC	one cycle (95 °C, 3 min) → 40 cycles (95 °C, 30 s → 64 °C, 30 s → 68 °C, 30 s)	U09861	171	[28]
	R	GTCGTTGATGACGCGGAAAG				
HSP 90	F	GGAGGATCACTTGGCTGTCA	one cycle (95 °C, 3 min) → 40 cycles (95 °C, 10 s → 62 °C, 30 s → 72 °C, 30 s)	NM_001012670	177	[28]
	R	GGGATTAGCTCCTCGCAGTT				
β-actin ^(1)^	F	AGCAAGCAGGAGTACGATGAGT		NM_173979.3	239	[29]
	R	ATCCAACCGACTGCTGTCA				
β-actin ^(2)^	F	CAGCAGATGTGGATCAGCAAGC		NM_173979.3	91	[27]
	R	AAC GCA GCT AAC AGT CCG CC				

^(1)^ Used as a housekeeping gene for IL-1B, IL-6, IL-10, HSP 70, and HSP 90. ^(2)^ Used as a housekeeping gene for TNF-a. IL-1B, interleukin-1β; IL-6, interleukin 6; IL-10, interleukin 10; TNF-a, tumor necrosis factor-α; HSP 70, heat shock proteins 70; HSP 90, heat shock proteins 90.

**Table 4 vetsci-09-00220-t004:** *In vitro* fermentation characteristics of the experimental diets after 48 h of incubation.

Items	CON	GINSENG	SEM	*p*-Value
Rumen parameters				
IVDMD (%)	85.6	88.1	0.52	0.006
IVNDFD (%aNDF)	66.0	72.9	1.19	0.002
IVCPD (%CP)	91.3	96.8	0.50	<0.001
pH	6.34	6.24	0.010	0.001
TVFA (mM)	81.2	87.5	4.34	0.227
Acetate (mmol/mol)	508.4	510.0	3.35	0.685
Propionate (mmol/mol)	302.3	302.2	2.33	0.678
Butyrate (mmol/mol)	138.3	135.9	1.42	0.256
A:P ratio	1.68	1.70	0.024	0.577
NH_3_-N (mg/dL)	35.2	37.6	0.68	0.043
Gas (mL/g DM)				
3 h	26.7	26.9	0.91	0.876
6 h	64.3	66.3	2.36	0.472
12 h	124.6	125.5	4.13	0.878
24 h	209.4	212.0	2.87	0.519
36 h	254.2	259.0	2.68	0.208
48 h	282.4	289.6	2.26	0.057
Fitted parameters of gas ^(1)^				
*K_g_*	0.038	0.038	0.0003	1.000
*V_max_*	341.7	353.8	7.14	0.210

SEM, standard error of the mean; DM, dry matter; IVDMD, dry matter digestibility; IVNDFD, neutral detergent fiber digestibility; IVCPD, crude protein digestibility; TVFA, total volatile fatty acids; and A:P ratio, acetate-to-propionate ratio; NH_3_-N, ammonia nitrogen ^(1)^ *K_g_*, fractional rate of gas production (h^−1^); *V_max_*, theoretical maximum gas production (mL/g DM).

**Table 5 vetsci-09-00220-t005:** Growth performance, respiration rate, and rectal temperature of heat-stressed Hanwoo steers supplemented with red ginseng byproduct.

Items	CON	GINSENG	SEM	*p*-Value
Initial BW (kg)	563.0	581.1	13.96	0.310
Final BW (kg)	582.6	599.3	14.74	0.379
DMI (kg/d)	7.82	7.80	0.326	0.961
ADG (g/d)	670.4	595.8	99.20	0.614
FCR	17.7	13.5	5.17	0.427
Respiration rate (breaths/min)	52.2	51.5	2.93	0.800
Rectal temperature (°C)	39.0	38.9	0.14	0.490

BW, body weight; DMI, dry matter intake; ADG, average daily gain; FCR, feed conversion ratio; SEM, standard error of the mean.

**Table 6 vetsci-09-00220-t006:** Rumen fermentation characteristics of heat-stressed Hanwoo steers supplemented with red ginseng byproduct.

Items	CON	GINSENG	SEM	*p*-Value
NH_3_-N (mg/dL)	3.56 ^b^	6.14 ^a^	0.815	0.003
TVFA (mM)	65.7	66.1	3.03	0.884
Acetate (mmol/mol)	603.9	601.7	8.68	0.795
Propionate (mmol/mol)	217.2	216.3	6.39	0.893
Butyrate (mmol/mol)	141.3	144.8	6.16	0.574
A:P ratio	2.80	2.81	0.100	0.939
pH	6.71	6.78	0.080	0.401

NH_3_-N, ammonia nitrogen; TVFA, total volatile fatty acids; and A:P ratio, acetate-to-propionate ratio; SEM, standard error of the mean. ^a,b^ Values in the same row with different superscripts indicate significant differences between treatments (*p* < 0.05).

**Table 7 vetsci-09-00220-t007:** Blood serum metabolites of heat-stressed Hanwoo steers supplemented with red ginseng byproduct.

Items	CON	GINSENG	SEM	*p*-Value
TP (g/dL)	6.20	6.54	0.303	0.261
AST (U/L)	52.4 ^b^	70.0 ^a^	4.72	<0.001
ALT (U/L)	17.3	19.2	1.07	0.088
BUN (mg/dL)	9.94	11.28	0.480	0.013
Ca (mg/dL)	9.40	9.52	0.337	0.718
IP (mg/dL)	7.15	7.28	0.203	0.522
Mg (mg/dL)	2.25	2.32	0.056	0.241
T-Chol (mg/dL)	111.9	112.3	7.63	0.966
TG (mg/dL)	19.9	18.5	1.43	0.402
Glu (mg/dL)	69.4	67.7	2.19	0.454
Alb (g/dL)	3.23	3.24	0.118	0.924
Crea (mg/dL)	1.34	1.34	0.064	0.987

TP, total protein; AST, aspartate aminotransferase; ALT, alanine aminotransferase; BUN, blood urea nitrogen; Ca, calcium; IP, inorganic phosphorus; Mg, magnesium; T-Chol, total cholesterol; TG, triglyceride; Glu, glucose; Alb, albumin; and Crea, creatinine; SEM, standard error of the mean. ^a,b^ Values in the same row with different superscripts indicate significant differences between treatments (*p* < 0.05).

**Table 8 vetsci-09-00220-t008:** Maximum and minimum temperature, relative humidity and temperature–humidity index during the experiment on Hanwoo steers.

Items	Maximum	Minimum
Temperature (°C)	29.3 ± 3.35	20.6 ± 2.03
Relative humidity (%)	89.0 ± 8.48	55.5 ± 12.53
THI ^(1)^	83.1 ± 4.96	66.4 ± 2.93

^(1)^ THI, temperature–humidity index.

## Data Availability

The data presented in this study are available on request from the corresponding author.

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
