# Peer review of "Effects of Red Ginseng Byproducts on Rumen Fermentation, Growth Performance, Blood Metabolites, and mRNA Expression of Heat Shock Proteins in Heat-Stressed Fattening Hanwoo Steers"

_vetsci, 2022, doi:10.3390/vetsci9050220_

Round 1

Reviewer 1 Report

This is to my opinion an interesting study and it is well presented. If have only some minor issues that should be considered. Please find specific comments in the following:

  • very much abbreviations were used, which makes it difficult to follow; I would suggest reducing the abbreviations to those that are really necessary
  • L68, explain why you have chosen 2% replacement and not more or less
  • L71, explain abbreviation NASEM
  • Tab. 2, revise the footnote . . . what is 33,330,000.00?
  • Sec. 2.2 - in vitro experiment, please provide pH, redox potential and temperature of ruminal fluid/inoculum immediately before starting the experiment (this is important to evaluate starting conditions for microbial fermentation); how many runs were performed (replicates) and how many technical replicates were within a run? did you perform blank correction for gas production and metabolites (important, because ruminal fluid itself has residual microbial activity!)
  • Tab. 3, explain why max. and min. values have SD
  • L173-174, how often did you record RR?
  • Sec. 2.3 - in vivo trial, an ethical statement is missing!
  • L329, which effect may the boiling process have on nutrients - especially on proteins (Maillard product formation?) - and on bioactive compounds?
  • L399, is "damage" the correct word here?
  • Conclusion, please give a clear statement if you think RGB application is meaningful or not under HS conditions and if the 2% replacement level is appropriate!

Author Response

Thank you for your detailed comment. We highlighted all the changed things using yellow color in the manuscript. Please see the attachment. The rebuttal is on page 17.

Reviewer 2 Report

Return date: March 22, 2022.

Dear author,

Type of manuscript: Article.

Title: Effects of red ginseng byproducts on rumen fermentation, growth performance, blood metabolites, and mRNA expression of heat shock proteins in heat-stressed fattening Hanwoo steers.

The experimental design of this manuscript isn’t rigorous and needs to be revised again.

  1. Row 60: Heat stress can increase the metabolism of livestock and poultry, cause excessive accumulation of reactive oxygen species, and cause oxidative-antioxidant imbalance in the body, resulting in oxidative stress; and then cause oxidative damage to tissue cells, proteins and nucleic acids. It is recommended that the authors add some oxidative stress experiments. For example, long-term chronic heat stress caused by high temperature will consume a large amount of antioxidants in the body and damage the body's antioxidant defense system (gene mutation, protein denaturation, and antioxidant enzyme activity)
  2. Row 97: The cattle used for in vitro experiments and the cattle used for in vivo experiments seem to be different. Is there any species difference? Can it prove that the two are homologous?
  3. Rows 101~103: How to avoid bacterial contamination in rumen in vitro experiments, will it change the original flora composition?
  4. Row 152: The existence of effective substances in red ginseng byproduct is not indicated in the text.The economic difference between adding red ginseng byproduct to the feed and cooling the barn, and reducing heat stress, there are many more.
  5. Rows 153~154: The subjects mentioned only the breed, not the gender, there is a difference between bulls and cows.
  6. Rows 274~290: Why did the author only study the rumen? If the later stage is related to growth performance, it is also necessary to study the physiological function and health status of nutrient absorption in the intestinal tract.
  7. Please list more recent references.

Thank you and best regards.

Author Response

Thank you for your detailed comment. We highlighted all the changed things using green color in the manuscript. Please see the attachment. The rebuttal is on page 17. 

Round 2

Reviewer 2 Report

The author has added some content to the article. Some sentences in the article hope to make some changes. The more scientific description is required.